# *KRAS* Mutations Impact Clinical Outcome in Metastatic Non-Small Cell Lung Cancer

**DOI:** 10.3390/cancers14092063

**Published:** 2022-04-20

**Authors:** Ella A. Eklund, Clotilde Wiel, Henrik Fagman, Levent M. Akyürek, Sukanya Raghavan, Jan Nyman, Andreas Hallqvist, Volkan I. Sayin

**Affiliations:** 1Sahlgrenska Center for Cancer Research, Department of Surgery, Institute of Clinical Sciences, University of Gothenburg, 40530 Gothenburg, Sweden; ella.ang@gu.se (E.A.E.); clotilde.wiel@gu.se (C.W.); 2Wallenberg Centre for Molecular and Translational Medicine, University of Gothenburg, 40530 Gothenburg, Sweden; 3Department of Oncology, Sahlgrenska University Hospital, 41345 Gothenburg, Sweden; jan.nyman@vgregion.se (J.N.); andreas.hallqvist@vgregion.se (A.H.); 4Department of Laboratory Medicine, Institute of Biomedicine, University of Gothenburg, 40530 Gothenburg, Sweden; henrik.a.fagman@vgregion.se (H.F.); levent.akyurek@gu.se (L.M.A.); 5Department of Clinical Pathology, Sahlgrenska University Hospital, 41345 Gothenburg, Sweden; 6Department of Microbiology and Immunology, Institute for Biomedicine, Sahlgrenska Academy, University of Gothenburg, 40530 Gothenburg, Sweden; sukanya.raghavan@microbio.gu.se; 7Department of Oncology, Institute of Clinical Sciences, University of Gothenburg, 40530 Gothenburg, Sweden

**Keywords:** lung cancer, *KRAS*, chemotherapy, immunotherapy, biomarker

## Abstract

**Simple Summary:**

In this retrospective study including 580 patients with metastatic (Stage IV) non-small cell lung cancer, we investigated whether *KRAS* mutational status had any impact on clinical outcome. First, we analyzed overall survival of patients grouped based on absence (*KRAS*^WT^) or presence (*KRAS*^MUT^) of mutations in *KRAS*. Next, we assessed the effect of first-line therapies on both groups: platinum doublet chemotherapy (PT), the backbone treatment for most patients with metastatic non-small cell lung cancer, and immune checkpoint blockade (ICB) given to Stage IV patients with high PD-L1 expressing tumors. We found that *KRAS*^MUT^ patients had better response to ICB, but worse response to PT compared to *KRAS*^WT^ patients and that *KRAS*^WT^ patients with high PD-L1 expressing tumors responded better to PT than ICB. Our findings will have immediate clinical value, as *KRAS* mutations and PD-L1 expression are routinely assessed in most patients diagnosed with lung cancer.

**Abstract:**

There is an urgent need to identify new predictive biomarkers for treatment response to both platinum doublet chemotherapy (PT) and immune checkpoint blockade (ICB). Here, we evaluated whether treatment outcome could be affected by *KRAS* mutational status in patients with metastatic (Stage IV) non-small cell lung cancer (NSCLC). All consecutive patients molecularly assessed and diagnosed between 2016–2018 with Stage IV NSCLC in the region of West Sweden were included in this multi-center retrospective study. The primary study outcome was overall survival (OS). Out of 580 Stage IV NSCLC patients, 35.5% harbored an activating mutation in the *KRAS* gene (*KRAS*^MUT^). Compared to *KRAS* wild-type (*KRAS*^WT^), *KRAS*^MUT^ was a negative factor for OS (*p* = 0.014). On multivariate analysis, *KRAS*^MUT^ persisted as a negative factor for OS (HR 1.478, 95% CI 1.207–1.709, *p* < 0.001). When treated with first-line platinum doublet (*n* = 195), *KRAS*^MUT^ was a negative factor for survival (*p* = 0.018), with median OS of 9 months vs. *KRAS*^WT^ at 11 months. On multivariate analysis, *KRAS*^MUT^ persisted as a negative factor for OS (HR 1.564, 95% CI 1.124–2.177, *p* = 0.008). *KRAS*^MUT^ patients with high PD-L1 expression (PD-L1^high^) had better OS than PD-L1^high^
*KRAS*^WT^ patients (*p* = 0.036). In response to first-line ICB, *KRAS*^MUT^ patients had a significantly (*p* = 0.006) better outcome than *KRAS*^WT^ patients, with a median OS of 23 vs. 6 months. On multivariable Cox analysis, *KRAS*^MUT^ status was an independent prognostic factor for better OS (HR 0.349, 95% CI 0.148–0.822, *p* = 0.016). *kRAS* mutations are associated with better response to treatment with immune checkpoint blockade and worse response to platinum doublet chemotherapy as well as shorter general OS in Stage IV NSCLC.

## 1. Background

Non-small cell lung cancer (NSCLC) is the most commonly diagnosed cancer worldwide with 2.1 million new cases and 1.8 million deaths annually [1]. NSCLC can be treated effectively with local management of the primary tumor in early stages, but the 5-year survival for patients with advanced metastatic (Stage IV) NSCLC is below 10% [2,3]. NSCLC generally tends to have a high overall mutational burden mainly due to exposure to exogenous mutagens such as tobacco smoke but also air pollution, causing high genomic instability and inter-patient heterogeneity [4].

The treatment options for Stage IV NSCLC patients have become dependent on molecular profiling because of the introduction of small molecule kinase inhibitors (SMKIs) targeting activating mutations in *EGFR, ALK, BRAF, RET* and *ROS1* oncogenes [5]. Unfortunately, the majority of Stage IV NSCLC patients (in non-SE Asian populations) lack a targetable driving mutation, for which platinum-based chemotherapy doublets (PT) have been the only available treatment option until recently [4]. The introduction of immune checkpoint blockade (ICB) targeting programmed death ligand 1 (PD-L1) or programmed cell death 1 (PD-1) proteins led to impressive clinical results in a subset of patients [6,7,8]. Consequently, all Stage IV patients without any clinically actionable mutations are assessed for ICB, single or in combination with chemotherapy, as a first-line treatment option [9,10,11,12]. PD-L1 expression is the only validated predictive marker for response to immunotherapy. However, its accuracy as an individual prediction tool is not as good as initially assumed since on one hand, PD-L1 negative patients have been reported as ICB responders, while on the other, many patients with tumors expressing high levels of PD-L1 are being reported as non-responders to ICB therapy [9,13]. Along these lines, a meta-analysis of six randomized controlled trials with ICB showed that PD-L1 expression levels were neither prognostic nor predictive for OS [14].

The most frequent oncogenic driver in NSCLC is the Kirsten rat sarcoma viral oncogene (*KRAS*) present in up to 40% of all cases, and the most common mutations are G12C, G12V and G12D [15,16]. *KRAS* mutations have been considered to negatively influence the prognosis of NSCLC. Accordingly, *KRAS* mutations have been associated with a shorter OS following first-line PT treatment in more recent studies [17,18,19,20,21]. General attempts to target mutant KRAS and its downstream mediators in cancer therapy remain largely unsuccessful [4,22]. More recently, several inhibitors specifically binding mutant KRAS-G12C have been investigated in clinical trials; among these, Sotorasib is the first treatment to gain approval as second-line therapy for adults with NSCLC whose tumors have KRAS-G12C mutation [23,24,25,26,27]. However, like other SMKIs, a heterogeneous pattern of resistance to KRAS-G12C inhibition has already been observed [28,29]. Concurrently, it has been suggested that NSCLC harboring *KRAS* mutations might benefit from ICB therapy compared to *KRAS* wild-type tumors [30]. Nevertheless, retrospective real-world data have so far shown inconclusive results [30,31,32,33,34,35].

The gradual introduction of ICB treatment in recent years brought a new era with high hopes for patients with Stage IV NSCLC. Clinical trials have shown ICB as being superior to chemotherapy in patients harboring tumors with a high tumor proportion score (TPS) ≥ 50% for PD-L1. These results were confirmed in a recent network meta-analysis [7,8,10,36]. However, the impact of mutant *KRAS* was not addressed in these analyses. In addition, multiple real-world studies have reported conflicting results regarding the impact of mutant *KRAS* on ICB response. In the early days of immunotherapy, only PD-L1 ≥ 50% patients were included in first-line ICB treatment. We know now that PD-L1 negative patients may also respond to ICB [9,13]. Interestingly, some studies suggest that PD-L1 expression per se might have a prognostic impact on NSCLC [37,38].

Importantly, as of today and for the coming years, tumors from patients are being routinely assessed for only a few targetable genomic alterations in *KRAS*, *EGFR, ALK, BRAF, RET* and *ROS1* genes. Broad panel sequencing for guided treatment decisions is currently not clinically implemented, as the required infrastructure and expertise are, in most cases, still beyond clinical healthcare means. Thus, we ask whether *KRAS* mutational status could become a diagnostic biomarker to direct therapy choice.

By including all consecutive patients diagnosed with Stage IV NSCLC and molecularly assessed between 2016–2018 in West Sweden, the current retrospective cohort study provided a unique real-world dataset for assessing the impact of *KRAS* mutations and PD-L1 expression on OS following first-line standard of care, including platinum doublet chemotherapy and immunotherapy.

## 2. Materials and Methods

### 2.1. Patient Population

We conducted a multi-center retrospective study including all consecutive NSCLC patients diagnosed with Stage IV NSCLC and having molecular assessment performed between 2016–2018 in the Region Västra Götaland (region of West Sweden), Sweden (*n* = 580). Our study focused on patients diagnosed with Stage IV NSCLC, who, at the time, were the only NSCLC patients receiving first-line ICB therapy. During this period, patients diagnosed with squamous cell carcinoma were molecularly assessed to a lesser extent. Patient demographics (including age, gender, Eastern Cooperative Oncology Group (ECOG) performance status and smoking history), cancer stage, number of metastasis locations, pathological details (histology, mutation status including *KRAS* mutational status and subtype), first-line treatment and outcome data were retrospectively collected from patient charts and the Swedish Lung Cancer Registry. Approval from the Swedish Ethical Review Authority (Dnr 2019-04771) was obtained prior to study commencement.

### 2.2. Mutational Status

Patients were assessed with NGS for mutational status on DNA from FFPE blocks or cytological smears using the Ion AmpliSeq™ Colon and Lung Cancer Panel v2 from Thermo Fisher Scientific, Waltham, MA, USA, as a part of the diagnostic workup process at the Department of Clinical Pathology at Sahlgrenska University Hospital, Gothenburg, Sweden, assessing hotspot mutations in *EGFR, BRAF, KRAS* and *NRAS*. Until June 2017, *ALK* fusions were assessed with immunohistochemistry (IHC), and with fluorescence in situ hybridization (FISH) if positive or inconclusive IHC; ROS1 was analyzed upon request with FISH. Thereafter, *ALK, ROS1* and *RET* fusions were assessed on RNA using the Oncomine Solid Tumor Fusion Panel from Thermo Fisher Scientific.

### 2.3. PD-L1 Expression

Programmed death ligand 1 (PD-L1) expression was determined based on percentage of tumor cells with positive membranous staining and was reported as the tumor proportion score (TPS): PD-L1 negative TPS < 1%, low TPS 1–49%, high TPS ≥ 50%. PD-L1 expression was detected using the PD-L1 IHC 28-8 pharmDx system during routine diagnostic workup, and staining was assessed by lung pathologists.

### 2.4. ICB Treatment

During the time period of this study, the only ICB treatment approved for first-line treatment was Pembrolizumab, a humanized antibody targeting PD-1. The criteria were PD-L1^high^ TPS ≥ 50% for first-line and PD-L1^low^ TPS ≥ 1% for second-line treatment.

### 2.5. PT Treatment

Platinum doublet treatment consists of carboplatin or cisplatin in combination with one more non-platinum chemotherapy agent such as pemetrexed, vinorelbin, gemcitabine, paclitaxel, etoposide or vincristine.

### 2.6. Study Objectives

The primary outcome of this study was OS, defined as the interval between the date of diagnostic sample collection and the date of death from any cause. Patients alive or lost to follow-up at data collection were censored at last contact. Median follow-up time was 35 months (95% CI 31.1–38.9) and was estimated using the reverse Kaplan–Meier method.

We compared OS stratified on *KRAS*^WT^ and *KRAS*^MUT^ for the entire cohort, for all patients receiving life-extending treatment excluding patients receiving best supportive care or palliative radiotherapy (e.g., single tumor radiation for pain relief), and for all patients receiving PT or ICB as first-line treatment. We also investigated the impact of TPS score (negative, low and high) on OS, stratified on *KRAS* mutational status. Finally, we assessed the clinical impact of *KRAS*^G12C^ mutations compared to *KRAS*^non-G12C^ mutations for the entire cohort, including OS and response to first-line therapies.

### 2.7. Statistical Analysis

Clinical characteristics were summarized using descriptive statistics and evaluated with univariate analysis in table form. Survival was estimated using the Kaplan–Meier method. Log-rank test was used to assess significant differences in OS between groups. Multivariable Cox regression analyses were conducted to compensate for potential confounders. Statistical significance was set at *p* < 0.05, and no adjustments were made for multiple comparisons. One-sided Fisher’s exact test was used to determine the connection between *KRAS* status and being alive in the ICB-treated group. Data analysis was conducted using IBM SPSS Statistics version 27 and GraphPad Prism version 9.

## 3. Results

### 3.1. Patients and Tumor Characteristics

A total of 597 consecutive patients diagnosed with Stage IV NSCLC were molecularly assessed during 2016–2018 in West Sweden. A total of 17 patients were excluded for having incomplete medical records, for receiving palliative treatment before 2016, for incorrect diagnosis or for receiving simultaneous treatment for another type of cancer (Figure 1A). Among the 580 included patients, more than a third harbored a *KRAS* mutation (206, 35.5%), the majority were female (326, 56.2%), the median age was 71 years, and 80% were current or former smokers (Table 1). The majority of patients (322, 55.5%) had a good performance status (PS) with low ECOG 0–1 at diagnosis (Table 1). Histologically, the vast majority of NSCLCs were adenocarcinoma of the lung (498, 85.9%), while squamous cell carcinoma cases were relatively low, which was expected due to the selection of histological type for NGS assessment (32, 5.5%) (Table 1). In line with earlier studies, patients diagnosed with a low ECOG score had a significantly better overall survival (OS) compared to patients diagnosed with a higher ECOG score (*p* = 0.0001) (Appendix A) [3,39]. In addition, higher numbers of metastasis locations at diagnosis correlated with poor OS (*p* = 0.0002) (Appendix A), which was expected [3,39]. Furthermore, there was a trend towards better OS for females compared to males (*p* = 0.064) (Appendix A) [39,40]. When comparing the baseline characteristics of *KRAS*^WT^ with *KRAS*^MUT^ patients, there were more females and a higher proportion of current and former smokers in the *KRAS*^MUT^ population (Table 1).

### 3.2. KRAS Mutations Are a Negative Factor for Overall Survival in Stage IV NSCLC

Survival estimates for the whole study cohort displayed that *KRAS* mutations were a significant negative factor for overall survival (*p* = 0.014) (Figure 1B). On multivariate Cox regression analysis, *KRAS*^MUT^ status and ECOG 2 or higher were independent prognostic factors for worse OS (HR 1.478, 95% CI 1.207–1.709, *p* < 0.001 and HR 3.175, 95% CI 2.622–3.845, *p* = 0.038). When comparing *KRAS*^G12C^ and *KRAS*^non-G12C^, no difference in OS was observed (*p* = 0.222) (Appendix A). The population that received any type of life-extending treatment (*n* = 388), excluding patients only receiving best supportive care or palliative radiotherapy (e.g., single tumor radiation for pain relief), did not show a significant difference in OS between KRAS^MUT^ and KRAS^WT^ (*p* = 0.095) (Figure 1C). Similarly, when comparing *KRAS*^G12C^ and *KRAS*^non-G12C^, no difference in OS was observed among treated patients (*p* = 0.250) (Appendix A).

When treated with first-line PT, one of the standard treatments at the time (*n* = 219), KRAS^MUT^ was a significant negative factor for survival (*p* = 0.001), with median OS of 9 months vs. *KRAS*^WT^ of 14 months (Appendix A). Baseline characteristics of the patients showed that around 20% of *KRAS*^WT^ patients displayed druggable genomic alterations in *ALK* or *EGFR* (Appendix A). To eliminate the risk of these patients driving any difference between *KRAS*^MUT^ and *KRAS*^WT^ groups due to potentially receiving targeted therapies in the second line of treatment, we excluded all patients with EGFR mutations and ALK fusions in the *KRAS*^WT^ group from further analysis. Nevertheless, in patients receiving first-line PT chemotherapy (*n* = 195), *KRAS*^MUT^ was a significant (*p* = 0.018) negative factor for survival (Figure 2A), with median OS of 9 months for *KRAS*^MUT^ vs. 11 months for *KRAS*^WT^. In multivariate Cox regression analysis, *KRAS*^MUT^ status and ECOG 2 or higher were independent prognostic factors for worse OS (HR 1.564, 95% CI 1.124–2.177, *p* = 0.008 and HR 1.906, 95% CI 1.307–7779, *p* < 0.001). When comparing *KRAS*^G12C^ and *KRAS*^non-G12C^, there was no difference in OS (*p* = 0.313) (Figure 2B).

### 3.3. Overall Survival for Stage IV NSCLC Patients Treated with First-Line PT or ICB Stratified Based on KRAS Mutational Status

When comparing OS for all Stage IV patients receiving PT (*n* = 195) or ICB (*n* = 37) in a first-line setting, there was no difference between the two treatment groups (*p* = 0.678), with median OS of 11 months for PT vs. 13 months for ICB (Figure 3A). *KRAS*^MUT^ patients treated with ICB (*n* = 20) had a significantly (*p* = 0.003) better outcome compared to patients treated with PT (*n* = 104), with median OS of 23 vs. 9 months (Figure 3B). *KRAS*^WT^ patients treated with ICB (*n* = 17) had significantly (*p* = 0.023) worse survival than patients treated with PT (*n* = 91) with a median OS of 6 vs. 11 months (Figure 3C).

### 3.4. PD-L1 Expression in Tumors from Stage IV Patients Is a Positive Factor for Overall Survival in KRAS^MUT^ but Not KRAS^WT^ Patients

Currently, tumoral PD-L1 expression is the only biomarker for ICB treatment widely used in the clinic for NSCLC [41,42]. When analyzing all treated Stage IV NSCLC patients with known PD-L1 status (*n* = 261) stratified on *KRAS* mutational status and PD-L1 expression levels (negative, low or high) (Figure 4A), we observed that increased PD-L1 expression correlated with significantly better OS in *KRAS*^MUT^ patients (*p* = 0.036, Figure 4B). *KRAS*^MUT^ patients were clearly separated on OS between PD-L1 negative, low and high groups with median OS of 6, 11 and 17 months, respectively (Figure 4B). In multivariate Cox regression analysis, PD-L1 was an independent positive factor, and ECOG 2 or higher were independent factors for worse OS (HR 0.993 95% CI 0.986–0.999, *p* = 0.028 and HR 2.470, 95% CI 1.470–4.273, *p* = 0.001) for *KRAS*^MUT^ patients (Appendix A). No correlation between PD-L1 expression and OS in *KRAS*^WT^ patients was observed (Figure 4C). No factor was significant in multivariate analysis (Appendix A). There was a larger proportion of PD-L1^high^ in the *KRAS*^MUT^ population compared to *KRAS*^WT^ (43.0% vs. 32.7%).

### 3.5. KRAS Mutations Are a Positive Factor for Overall Survival in Patients with PD-L1^high^ Tumors Receiving Immunotherapy

We next analyzed the specific impact of PD-L1^high^ expression on OS, comparing all PT-treated vs. ICB-treated patients. Outcome was significantly improved for *KRAS*^MUT^ patients on ICB treatment (*p* = 0.028), with median OS of 9 vs. 23 months (Figure 5A). *KRAS*^WT^ PD-L1^high^ patients displayed a worse outcome among patients on ICB treatment compared to those on PT treatment (*p* = 0.010), with median OS of 6 months vs. 28 months for each respective treatment (Figure 5B). Importantly, the *KRAS*^MUT^ group had a significantly (*p* = 0.006) better outcome on ICB treatment compared to the *KRAS*^WT^ group, with a median OS of 23 vs. 6 months (Figure 5C, Appendix A). On multivariate Cox regression analysis, *KRAS*^MUT^ status and ECOG PS 0–1 were independent prognostic factors for better OS following first-line ICB (HR 0.349, 95% CI 0.148–0.822, *p* = 0.016 and HR 0.398, 95% CI 0.165–0.963, *p* = 0.041). Finally, when looking at all patients treated with ICB individually (*n* = 37), there were significantly more *KRAS*^MUT^ patients alive at last follow-up compared to *KRAS*^WT^ patients (50% vs. 11.8%, *p* = 0.0152) (Figure 5D).

## 4. Discussion

The predictive value of *KRAS* mutations for survival in Stage IV NSCLC patients is still not clearly defined. While some studies have reported *KRAS* mutations to be a negative factor for survival estimates [3,22,43], others have shown this not to be the case [44]. The current retrospective cohort study showed a clear difference in OS for Stage IV NSCLC patients when stratified on *KRAS* mutational status. Previous studies have shown that *KRAS* mutations were associated with a shorter OS in response to chemotherapy [18,19]. In line with these studies, our data showed that *KRAS*^MUT^ is a potentially negative predictive factor for PT treatment. This is of importance, especially in light of the new treatment options now available, such as immunotherapy. Indeed, when stratifying Stage IV NSCLC patients receiving first-line ICB on *KRAS* mutational status, there was a clear increase in survival for *KRAS*^MUT^ patients.

Translational studies have suggested that *KRAS*^MUT^ patients might respond well to ICB due to a larger proportion of smokers and a higher immunologically active tumor environment as a consequence of constitutive activation of *KRAS* and downstream signaling pathways [30,31,32,33,34,39,45,46]. In agreement, our data showed that first-line ICB led to a clinically meaningful improvement in OS for *KRAS*^MUT^ patients when compared to first-line platinum doublet treatment. Furthermore, PD-L1 status had a clear and significant impact on stratifying OS in the *KRAS*^MUT^ group receiving treatment, whereas no difference was observed for the *KRAS*^WT^ group, which is in line with a previous study reporting a similar trend for *KRAS*^MUT^ NSCLC patients receiving ICB treatment [35].

Interestingly, *KRAS*^WT^ patients showed a better response to PT treatment than ICB. This might be explained by *KRAS*^WT^ being a heterogeneous group, with the majority of patients not having an identified driving mutation, especially as patients with targetable alterations in ALK and EGFR were excluded from further analysis.

There are several retrospective studies showing different outcomes when comparing *KRAS*^MUT^ and *KRAS*^WT^ response to ICB. These discrepancies could be explained by major study design differences, such as inclusion of several different stages at diagnosis, multiple treatment lines considered, and inclusion of patients with no known PD-L1 status [30,31,32,33,34,35,39]. A recent registry study from the Netherlands reported no difference in OS for PD-L1^high^ patients receiving first-line ICB monotherapy, but the fraction of patients where *KRAS* mutations status had been assessed and reported to the registry was uncertain and could only be approximated (at around 75%) [39]. However, a meta-analysis of randomized controlled trials showed a significant OS benefit for *KRAS*^MUT^ patients in first-line immunotherapy with or without chemotherapy vs. chemotherapy alone [46], which is in line with the current study. Among these, the retrospective analysis of the phase 3 study KEYNOTE042 indicated a clear trend towards better OS in the *KRAS*^MUT^ group when comparing first-line ICB versus platinum-containing chemotherapy (28 vs. 11 months) than for the patients with *KRAS*^WT^ (15 vs. 12 months) [45], albeit only 301 out of 782 patients were molecularly assessed for *KRAS* status, among which 69 *KRAS*^MUT^ cases were identified. Importantly, KEYNOTE042 did not address the impact of *KRAS* mutations on PT treatment outcome. In our cohort, 580 patients were assessed, and 206 *KRAS*^MUT^ patients were identified. To the best of our knowledge, our retrospective cohort study is the largest to date comparing response to first-line ICB and PT therapy in Stage IV NSCLC patients where the *KRAS* mutational status was known for all patients.

Currently, first-line ICB is an option for all NSCLC patients with PD-L1^high^ tumors. However, only a subset of patients benefit from durable response to immunotherapy, which does not correlate with tumoral PD-L1 expression. In line with other studies [9,13,14], our findings suggest that additional biomarkers for ICB response are clearly needed. Indeed, *KRAS*^WT^ patients with PD-L1^high^ tumoral expression responded better to PT than to ICB. In contrast, *KRAS*^MUT^ patients clearly benefited more from ICB treatment (50% alive at last follow-up). Our results suggest that tumoral PD-L1^high^ expression in combination with *KRAS* mutations is a better predictive biomarker for ICB in Stage IV NSCLC than PD-L1^high^ expression alone. Our findings will have immediate clinical value, as assessing both *KRAS* mutational status and PD-L1 expression is currently performed routinely for most NSCLC patents.

Although this study provides real-world evidence of the impact of *KRAS* mutational status on first-line therapy in a large and well-controlled group of Stage IV NSCLC patients, it is obviously limited by its retrospective nature. Despite the fact that all Stage IV NSCLC patients molecularly assessed in the region of West Sweden between 2016–2018 were included in this cohort study, a relatively small number of patients were treated with first-line ICB therapy when compared to first-line PT therapy. Hence, pooled analyses from multi-cohort studies will be important to validate and expand our findings.

## 5. Conclusions

Here we reported that *KRAS* mutations are associated with a better response to treatment with immunotherapy and worse response to platinum doublet chemotherapy as well as shorter general overall survival in Stage IV NSCLC. Our findings suggest that *KRAS* mutations combined with PD-L1^high^ expression were associated with better response to first-line immunotherapy than PD-L1^high^ alone in patients with Stage IV NSCLC.

## Figures and Tables

**Figure 1 cancers-14-02063-f001:**
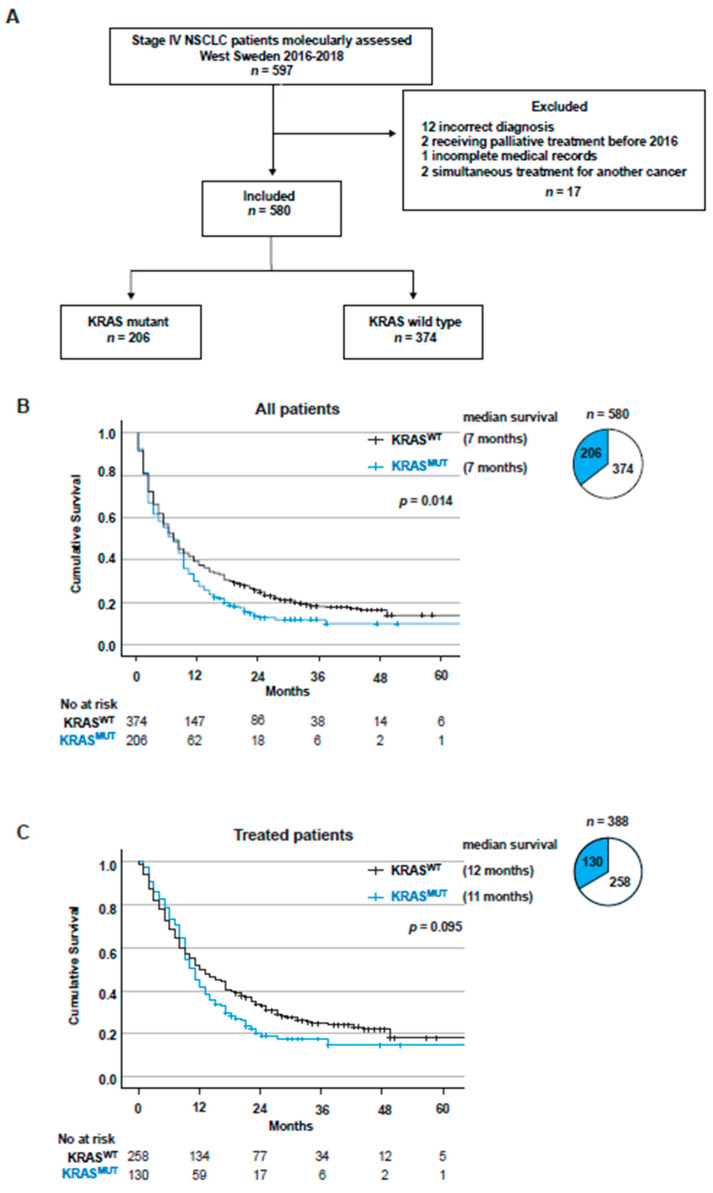
*KRAS* mutations are a negative factor for overall survival. (**A**) Flow chart showing the patient selection for the study. (**B**) Kaplan–Meier estimates comparing overall survival and median survival stratified on *KRAS*^WT^ vs. *KRAS*^MUT^ for the full cohort (*n* = 580). Pie chart showing patient distribution between *KRAS*^WT^ and *KRAS*^MUT^. (**C**) Kaplan–Meier estimates comparing overall survival for all receiving life-extending treatment (no treatment and only palliative radiotherapy excluded) stratified on *KRAS*^WT^ vs. *KRAS*^MUT^. Pie chart showing patient distribution between *KRAS*^WT^ and *KRAS*^MUT^.

**Figure 2 cancers-14-02063-f002:**
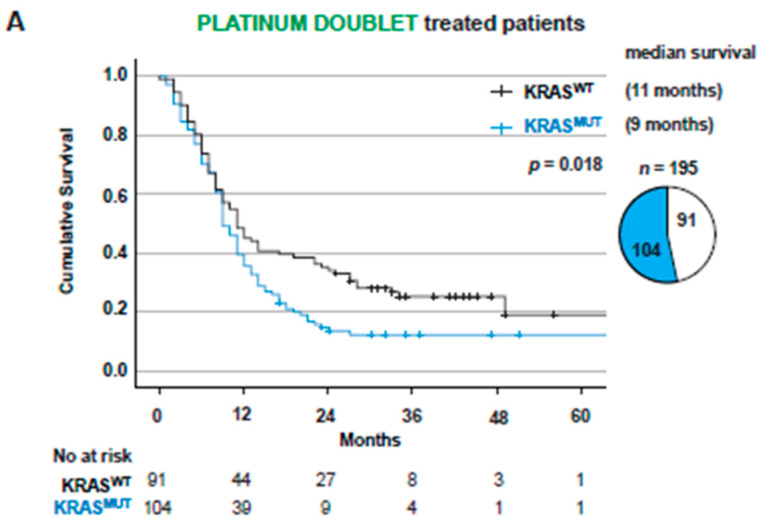
*KRAS*^MUT^ is a negative factor for overall survival when treated with platinum doublet independently of EGFR and ALK mutations. (**A**) Kaplan–Meier estimates comparing overall survival for all receiving treatment with platinum doublet stratified on *KRAS*^WT^ vs. *KRAS*^MUT^. Patients with EGFR and ALK alterations were excluded from *KRAS*^WT^ group. Pie chart showing patient distribution between *KRAS*^WT^ and *KRAS*^MUT^. (**B**) Kaplan–Meier estimates comparing overall survival for all *KRAS*^MUT^ receiving treatment with platinum doublet stratified on *KRAS*^G12C^ vs. *KRAS*^non G12C^. Pie chart showing patient distribution between *KRAS*^G12C^ and *KRAS*^non G12C^.

**Figure 3 cancers-14-02063-f003:**
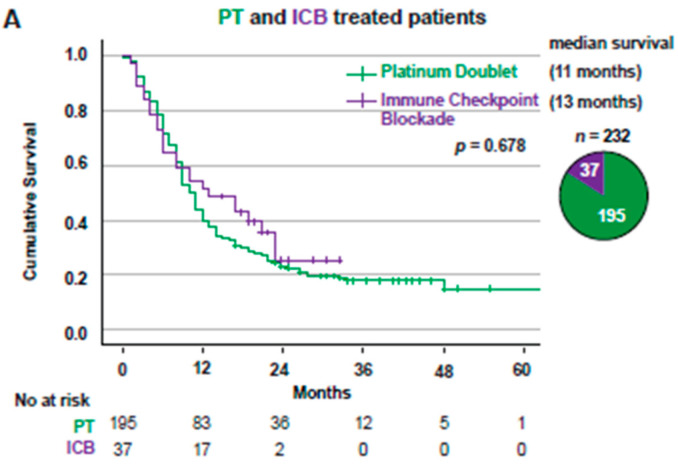
*KRAS*^MUT^, but not *KRAS*^WT^, had a better outcome with ICB than with platinum doublet treatment. (**A**) Kaplan–Meier estimates comparing overall survival for all patients, excluding patients with ALK or EGFR alterations, receiving treatment with PT vs. ICB. Pie chart showing patient distribution between PT and ICB. (**B**) Kaplan–Meier estimates comparing overall survival for all *KRAS*^MUT^ receiving treatment with PT vs. ICB. Pie chart showing patient distribution between platinum doublet and ICB. (**C**) Kaplan–Meier estimates comparing overall survival for *KRAS*^WT^, excluding patients with ALK or EGFR alterations, receiving treatment with platinum doublet vs. ICB. Pie chart showing patient distribution between PT and ICB. ICB: Immune checkpoint blockade; PT: Platinum doublet.

**Figure 4 cancers-14-02063-f004:**
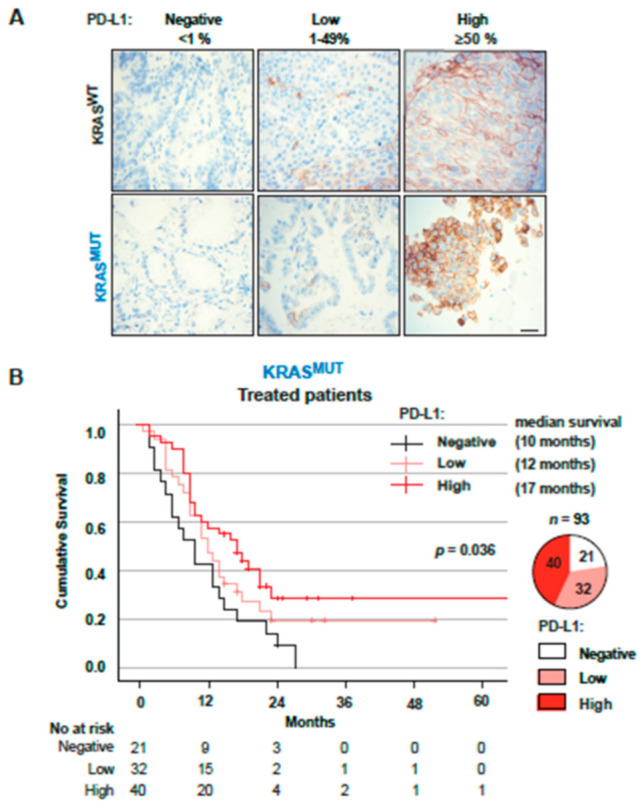
PD-L1 status has an impact on overall survival for treated patients with *KRAS*^MUT^**.** (**A**) IHC depicting examples of PD-L1 staining used to assess PD-L1 status in both *KRAS*^WT^ and *KRAS*^MUT^ NSCLC patients. PD-L1 expression is classified as negative, <50% or ≥50% based on tumor proportion score. (**B**) Kaplan–Meier estimates comparing overall survival for *KRAS*^MUT^ treated patients stratified on PD-L1 status. Pie chart showing patient distribution between PD-L1 negative, <50% and ≥50%. (**C**) Kaplan–Meier estimates comparing overall survival for *KRAS*^WT^ treated patients stratified on PD-L1 status. Pie chart showing patient distribution between PD-L1 negative, <50% and ≥50%. IHC: Immunohistochemistry; PD-L1: Programmed death ligand 1. Scale bar: 100 μm.

**Figure 5 cancers-14-02063-f005:**
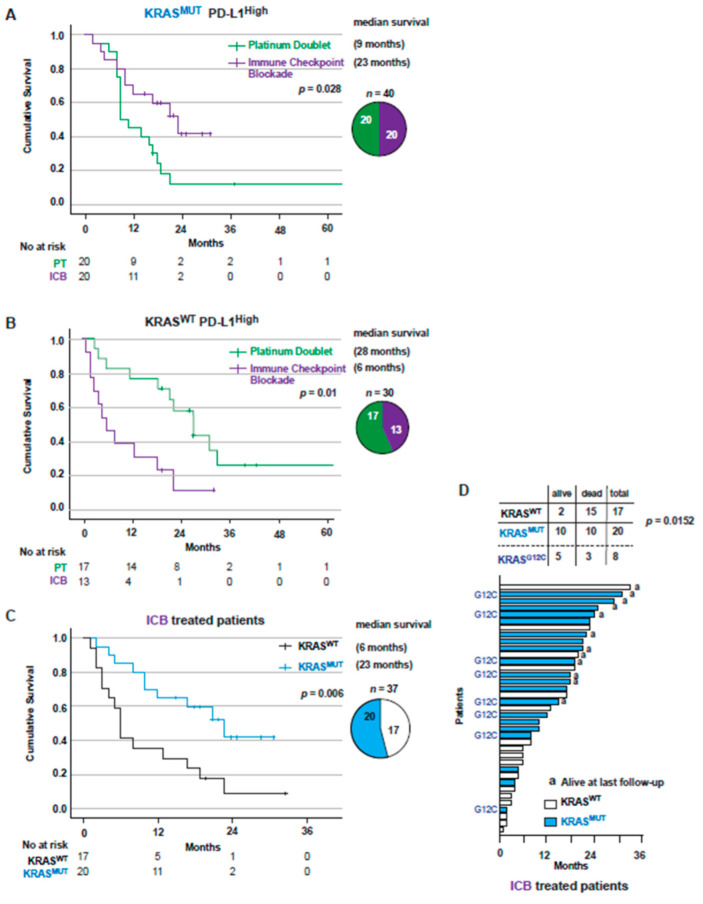
*KRAS*^MUT^ patients have a better outcome on ICB treatment than *KRAS*^WT^ patients. (**A**) Kaplan–Meier estimates comparing overall survival for *KRAS*^MUT^ PD-L1 ≥ 50% treated patients receiving PD or ICB. Pie chart showing patient distribution receiving PT or ICB treatment. (**B**) Kaplan–Meier estimates comparing overall survival for *KRAS*^WT^ PD-L1 ≥ 50% treated patients receiving PT or ICB. Pie chart showing patient distribution receiving PT or ICB treatment. (**C**) Kaplan–Meier estimates comparing overall survival for all patients receiving ICB treatment in a first-line setting. Pie chart showing patient distribution between *KRAS*^WT^ and *KRAS*^MUT^. (**D**) Swimmer plot showing survival time for both *KRAS*^WT^ and *KRAS*^MUT^ ICB-treated patients. Patients with KRAS^G12C^ mutation are marked with G12C. Status alive at last follow-up is shown. ICB: Immune checkpoint blockade; PT: Platinum doublet; PD-L1: Programmed death ligand 1.

**Table 1 cancers-14-02063-t001:** Characteristics of the total cohort as well as stratified on *KRAS*^WT^ and *KRAS*^MUT^. Data are presented as *n* (%).

Characteristics of All Patients Stage IV NSCLC	Total	KRAS WT	KRAS MUT
	*n* (%)	*n* (%)	*n* (%)
Total	580 (100)	374 (64, 5)	206 (35, 5)
Age in years, median (range)	71 (24–94)	70,5 (24–94)	71 (46–90)
Sex			
Male	254 (43, 8)	173 (46, 3)	81 (39, 3)
Female	326 (56, 2)	201 (53, 7)	125 (60, 7)
Smoking history			
Current smoker	192 (33, 1)	117 (31, 4)	75 (36, 4)
Former smoker	270 (46, 6)	152 (40, 6)	118 (57, 3)
Never smoker	116 (20)	103 (27, 5)	13 (6, 3)
Missing	2 (0, 3)	2 (0, 5)	0
Performance status			
ECOG 0	79 (13, 6)	48 (12, 8)	31 (15, 0)
ECOG 1	243 (41, 9)	142 (38, 0)	101 (49, 0)
ECOG 2	154 (26, 6)	99 (26, 5)	55 (26, 7)
ECOG 3	74 (12, 8)	57 (15, 2)	17 (8, 3)
ECOG 4	19 (3, 3)	17 (4, 5)	2 (1, 0)
Missing	11 (1, 9)	11 (2, 9)	0
Histology			
Adenocarcinoma	498 (85, 9)	316 (84, 5)	128 (85, 9)
NSCLC NOS	50 (8, 6)	28 (7, 5)	19 (12, 8)
Squamous cell carcinoma	32 (5, 5)	30 (8, 0)	2 (1, 3)
Mutation status			
None known	231 (39, 8)	231 (71, 8)	
ALK	19 (3, 3)	19 (5, 1)	
EGFR	85 (14, 7)	85 (22, 7)	
BRAF	20 (3, 4)	20 (5, 3)	
Other	15 (2, 6)	15 (4)	
ROS1	2 (0, 3)	2 (0, 5)	
RET	2 (0, 3)	2 (0, 5)	
KRAS	206 (35, 5)		
KRAS submutation			
G12A			14 (6, 8)
G12C			83 (40, 3)
G12D			23 (11, 2)
G12V			43 (20, 9)
Q61H			10 (4, 9)
Others			33 (16)
At last follow-up			
Alive	97 (16, 7)	69 (18, 4)	28 (13, 6)
Deceased	483 (83, 3)	305 (81, 6)	178 (86, 4)
Survival			
Median survival (months)	7	7	7
No. of metastatic locations at diagnosis			
1	361 (62, 2)	234 (62, 6)	127 (61, 7)
2	153 (26, 4)	101 (27, 0)	52 (25, 2)
3	47 (8, 1)	30 (8, 0	17 (8, 3)
>3	11 (1, 9)	3 (0, 8)	8 (3, 9)
Missing	8 (1, 4)	6 (1, 6)	2 (1, 0)

ECOG PS, Eastern Cooperative Oncology Group Performance Status.

## Data Availability

Data from this study are available upon reasonable request.

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
