# Peer review of "KRAS Mutations Impact Clinical Outcome in Metastatic Non-Small Cell Lung Cancer"

_cancers, 2022, doi:10.3390/cancers14092063_

Round 1

Reviewer 1 Report

The authors assessed KRAS mutation status in patients with metastatic (stage IV) non-small cell lung cancer (NSCLC) as a positive predictor of immune checkpoint blockade therapy and a negative predictor of platinum-based two-agent chemotherapy and general OS in stage IV NSCLC. This manuscript is interesting and clinically informative. The reviewer has the following concerns to prompt the manuscript for publication.

The introduction describes the study background well, however, "two inhibitors efficiently targeting mutant KRAS-G12C have been developed but the clinical efficacy is yet to be determined in ongoing trials [23-27]."
This statement is not a particularly appropriate summary, as it is important to note that the KRAS G12C inhibitor AMG 510 has been approved for clinical use. Despite the fact that potential resistance to this class of molecules has emerged (PMID: 34864132). Hence, the authors should clearly describe here the KRAS G12C inhibitors that are already available for clinical use and point out that possible resistance also exists. These indications and citations are necessary.
The necessary clinical ethics approvals are required to be uploaded.
Univariate and multivariate analysis methods need to be progressively clarified. The graphs should preferably be presented in vector form. Figure 4A is vague and therefore the IHC results are unclear.

Reviewer 2 Report

In this article, Eklund et al. present a retrospective study of KRAS mutations in a large cohort of advanced lung adenocarcinoma patients from several Swedish hospitals. The possible predictive or prognostic utility of KRAS mutations in NSCLC has been the subject of several publications and it is still controversial. At this respect, the manuscript is not novel. However, the number of patients analyzed is high and includes patients treated not only with chemotherapy but also with immune checkpoint therapy (ICT). Therefore, the manuscript makes a significant contribution and deserves to be published after a somewhat extensive revision

Major comments

  1. Some studies have found that G12C and non-G12C mutations might have a different clinical impact of in advanced NSCLC (i.e.; Aredo et al, Lung Cancer, 2019; Tao et al, Clin Lung Cancer, 2012; Molina-Vila et al, CCR, 2014). Also, the largest study published so far in early stage NSCLC reached the same conclusion (Finn et al, 2021). The authors should study separately in their Swedish cohort the clinical impact of G12C and non-G12C KRAS mutations and include the results in the article
  2. All over the article (also in the title), the authors talk about NSCLC. However, KRAS mutations are rare in squamous cell carcinoma, a histology that was practically excluded from the study, which was limited to adenocarcinoma (86% samples). The authors should modify the entire manuscript accordingly and make it clear that their conclusions basically apply to adenocarcinomas.
  3. The median follow-up of the study (7 months) is very short, particularly for studying OS, and might bias the results. The number of patients at risk should be included in all Kaplan-Meier plots presented in the article so that the reader can evaluate this possible bias. In addition, it’s the usual way of presenting Kaplan-Meier’s in scientific literature.
  4. In the abstract and conclusion sections, the authors claim that KRAS mutations are predictive factors for ICT and chemotherapy. However, the evidence presented is not sufficient to sustain this claim; a validation cohort and ideally a clinical trial would be needed. The authors should replace the claim by something like “KRAS mutations were associated with better (or worse) response to…”
  5. NSCLC is usually classified in surgical (Stages I-IIIA) and advanced (Stages IIIB-IV) and a vast majority of studies use this classification. However, the authors only included Stage IV and not Stage IIIB patients in their analysis. The rationale for this unusual selection should be explained

Minor comments

-PD is universally used as the abbreviation for progressive disease, not for platinum doublet chemotherapy, which is usually abbreviated PT, PtC or PBDC

-Line 50. This is only true in non-SE Asian populations. In SE Asia prevalence of EGFR-mut alone is almost 50%

-Line 54. Results of ICB are only “impressive” in the 20-30% of NSCLC patients responding. The sentence should be modified

-Table 1. The authors mention in methods that all tumors where an adenocarcinoma component could not be ruled out were included in the study. They should clarify if the squamous cell carcinomas that appear in Table 1 were adenosquamous (particularly those with KRAS mutations)

-Pages 181-265. The authors should add (as supplementary Tables) the results of the multivariate analyses they performed so that the reader knows what factors were included

-Lines 230-241. A multivariate Cox analysis should be performed in both groups of patients (KRAS mut and KRAS wt)

-Lines 253-265, Fig 5. The number of patients analyzed is very small and the results should be taken with caution. The authors should make this point clear in the text. Also, the type of plot in Fig 5D (waterfall) is universally used to show changes in tumor size. It should be removed and replaced by the patient’s at risk numbers, as mentioned above

-The sentence in lines 323-324 should be re-written

Round 2

Reviewer 1 Report

All concerns have been addressed by the authors.

Reviewer 2 Report

All my previous concerns have been satisfactorly addressed by the authors